# Gram-Negative Bacteria Holding Together in a Biofilm: The *Acinetobacter baumannii* Way

**DOI:** 10.3390/microorganisms9071353

**Published:** 2021-06-22

**Authors:** Arianna Pompilio, Daniela Scribano, Meysam Sarshar, Giovanni Di Bonaventura, Anna Teresa Palamara, Cecilia Ambrosi

**Affiliations:** 1Center for Advanced Studies and Technology (CAST), Department of Medical, Oral and Biotechnological Sciences, Service of Clinical Microbiology, “G. d’Annunzio” University of Chieti-Pescara, 66100 Chieti, Italy; arianna.pompilio@unich.it (A.P.); gdibonaventura@unich.it (G.D.B.); 2Department of Public Health and Infectious Diseases, Sapienza University of Rome, 00185 Rome, Italy; daniela.scribano@uniroma1.it; 3Dani Di Giò Foundation-Onlus, 00193 Rome, Italy; 4Research Laboratories, Bambino Gesù Children’s Hospital, IRCCS, 00146 Rome, Italy; meysam.sarshar@uniroma1.it; 5Department of Infectious Diseases, Istituto Superiore di Sanità, 00161 Rome, Italy; annateresa.palamara@uniroma1.it; 6Laboratory Affiliated to Institute Pasteur Italia-Cenci Bolognetti Foundation, Department of Public Health and Infectious Diseases, Sapienza University of Rome, 00185 Rome, Italy; 7Department of Human Sciences and Promotion of the Quality of Life, San Raffaele Open University, IRCCS, 00166 Rome, Italy

**Keywords:** *Acinetobacter baumannii*, biofilm, multidrug-resistant, prevention, treatment

## Abstract

Bacterial biofilms are a serious public-health problem worldwide. In recent years, the rates of antibiotic-resistant Gram-negative bacteria associated with biofilm-forming activity have increased worrisomely, particularly among healthcare-associated pathogens. *Acinetobacter baumannii* is a critically opportunistic pathogen, due to the high rates of antibiotic resistant strains causing healthcare-acquired infections (HAIs). The clinical isolates of *A. baumannii* can form biofilms on both biotic and abiotic surfaces; hospital settings and medical devices are the ideal environments for *A. baumannii* biofilms, thereby representing the main source of patient infections. However, the paucity of therapeutic options poses major concerns for human health infections caused by *A. baumannii* strains. The increasing number of multidrug-resistant *A. baumannii* biofilm-forming isolates in association with the limited number of biofilm-eradicating treatments intensify the need for effective antibiofilm approaches. This review discusses the mechanisms used by this opportunistic pathogen to form biofilms, describes their clinical impact, and summarizes the current and emerging treatment options available, both to prevent their formation and to disrupt preformed *A. baumannii* biofilms.

## 1. Introduction

Bacteria are fascinating microscopic cells that can live by themselves or be extremely social. Indeed, they can establish social interactions with other microorganisms to form highly organized communities known as biofilms. These consortia consist of adherent aggregates of microorganisms arranged within a matrix of a self-produced extracellular polymeric substance (EPS) composed of a mixture of polysaccharides, proteins and extracellular DNA [1].

Biofilm-forming activity is a widespread bacterial feature found on natural and artificial surfaces [2]. In natural environments, biofilms represent the preeminent lifestyle of bacteria which can have beneficial effects on plant growth promotion [3], organic compound degradation [1], including different aquatic ecosystems [4,5]. Moreover, microbial biofilms have been found useful in food fermentation, the production of many bio-based materials, bioremediation, wastewater treatment and microbial fuel cells [6,7,8,9,10]. 

Despite these beneficial and useful roles, biofilms also represent a significant threat for public health, being responsible for persistent infections with relevant economic and health impacts. In fact, biofilms account for 65% of microbial diseases and about 80% of chronic infections, associated to both medical devices and biotic surfaces [11]. Indeed, biofilms can occur on any type of medical device surface, such as catheters, breast implants, contact lenses, heart valves, pacemakers and defibrillators, ventricular shunts, and joint prostheses [12]. On the other hand, non-device-associated biofilms are represented by periodontitis, osteomyelitis, otitis media, biliary tract infection, and endocarditis [13]. Bacterial biofilm infections are very difficult to treat because these social interactions are highly resistant to antibiotic treatment and immune responses. The main pathogenic biofilm producers include the Gram-positive *Staphylococcus aureus*, *Staphylococcus epidermidis*, *Enterococcus faecalis*, *Streptococcus viridans* and the Gram-negative *Escherichia coli*, *Klebsiella pneumoniae*, *Acinetobacter baumannii*, *Proteus mirabilis*, and *Pseudomonas aeruginosa*. Among them, some belong to the group of most concerning antibiotic-resistant bacterial pathogens, named ESKAPEE (*Enterococcus faecium*, *S. aureus*, *K. pneumoniae*, *A. baumannii*, *P. aeruginosa*, *Enterobacter* spp. and *E. coli*). 

The poor susceptibility of biofilms to host immune defenses and antibiotics can be due to the barrier effect of the EPS, but also to substantial changes in gene expression that result in phenotypic, metabolic, and growth changes, including the genes involved in antibiotic resistance [14]. Although acting as a barrier, the EPS has a functional architecture to ensure the delivery of nutrients to all the cells embedded in the biofilm [15]. The architecture of biofilms depends on several factors, including the EPS composition, bacterial motility, intercellular communication, and environmental conditions. All these variables affect the development, shape and amount of biofilms that individual species can produce. In addition, despite several common strategies, Gram-negative bacteria have evolved unique features in biofilm formation because of their structural differences, compared to Gram-positive bacteria. In fact, Gram-negative bacteria are didermic cells in that their cell envelope contains an inner (IM) and an outer membrane (OM) and a periplasmic space in between. The peculiarities of their cell envelope and extracellular appendages play critical roles in biofilm formation.

*A. baumannii* has emerged as one of the most threatening members of the ESKAPEE group for having intrinsically or easily acquiring multiple antibiotic-resistance mechanisms and for its long-lasting persistence in the environment. Survival even in dried environmental conditions is tightly connected with its ability to form biofilm. This feature in conjunction with the overuse of antibiotics has allowed *A. baumannii* to perfectly adapt to healthcare environments, thereby representing a source of spread of this opportunistic pathogen. Therefore, the general aim of this review is to highlight improvements in the understanding of the biofilm formed by *A. baumannii*. The World Health Organization (WHO) listed this microorganism as a life-threatening bacterium due to its resistance to most, if not all, existing therapeutic treatments (https://www.who.int/news/item/ accessed on 22 June 2017). Specifically, the mechanisms underlying biofilm formation, the burden of antibiotic resistance, the clinical impact of the biofilm-related infections, and the therapeutic strategies for their treatment and prevention, are discussed.

## 2. Mechanisms of Biofilm Formation

### 2.1. Role of Fimbrial Structures

In recent years, the rates of antibiotic-resistant Gram-negative bacteria associated with biofilm-forming activity have increased worrisomely, particularly among healthcare-associated pathogens. In these microorganisms, biofilms offer additional protection against antibiotic treatment as well as biocides, disinfectants, host immune defenses, and desiccation, thereby allowing them to colonize and persist tenaciously on several surfaces. Biofilm formation is strongly induced when bacteria face suboptimal growth conditions or environmental stresses [16]. These stimuli trigger biofilm formation through complex regulatory mechanisms that involve global regulators, two-component and quorum sensing (QS) systems, cyclic nucleotides, nitric oxide, phenazines and small peptides [17,18].

The initial bacterial attachment is driven by the general physiochemical forces of the substrate and the bacterial cell envelope [11,17]. After the primary contact, the involvement of specific or nonspecific cell-surface adhesins trigger an irreversible attachment leading to the formation of microcolonies [19]. The attachment and proliferation of more cells generate more complex microcolony structures which start producing extracellular polysaccharides and other substances and, eventually, the final architecture of the biofilm [1]. In Gram-negative bacteria, cell-surface adhesion can be mediated by fimbrial and nonfimbrial adhesins as well as different polysaccharides. Fimbrial adhesins are proteinaceous extracellular fibers that play an important role in the initial bacterial adhesion to both biotic and abiotic surfaces, but also in later steps of biofilm development, such as intercellular aggregation and twitching motility [19,20,21,22]. These appendages can be classified in accordance to the pathways used by the bacteria for their secretion and export; thus, classical and nonclassical chaperone-usher pili are secreted by the type VII secretion system (T7SS), type IV pili uses the type II secretion system (T2SS), also called the general secretion pathway, while curli are secreted through the type VIII secretion system (T8SS), formerly known as the extracellular nucleation-precipitation pathway [19,23,24,25]. In addition, based on amino acid sequence comparisons of structural or assembly proteins, phylogenetic analyses have classified classical chaperone-usher pili (CUP), furthermore defined as class 1, 2 and 3, the nonclassical chaperone-usher pili, type IV pili as class 4 and curli as class 6. CUP are long, thin proteinaceous filaments that promote the formation of biofilms that mediate bacterial adhesion to abiotic surfaces and to other bacterial cells, as well as host−pathogen interactions [21,26,27,28]. The name comes from the pathway that enables the release of these pili onto the bacterial surface. From the cytoplasm, pilus subunits cross the IM through the Sec export system by a general secretory pathway (SecYEG machinery); once in the periplasm, a dedicated chaperone folds and stabilizes the pilus subunits [27]. Thus, the usher protein, a 24-stranded β-barrel OM pore, starts the polymerization and assists first the translocation of the most distal proteins, followed by the order of the growing pilus to the bacterial surface [27,29]. Some pili carry a tip adhesin at their distal end, a tip fibrillum, or an adaptor protein joining these structures [29]. A detailed description of the molecular mechanism of CUP biogenesis can be found in the review by Hospenthal et al. [27]. Besides *Enterobacteriaceae*, CUP were also found in several other Gram-negative bacteria, such as *A. baumannii*, *P. aeruginosa*, *Haemophilus influenzae*, *Burkholderia cenocepacia*, *Myxococcus xanthus*, *Gallibacterium* sp, *Rhizobium* sp. and *Xylella fastidiosa* where they mediate bacterial attachment and biofilm formation [30,31,32,33,34,35,36,37]. This large protein family was grouped into six clades (i.e., α, β γ, κ, π, and σ) based on the phylogenetic analysis of the amino acid sequences of the usher proteins [26]. The β clade is a small clade, whereas the γ clade is huger and quite heterogeneous, so that it was further subdivided into four subclades, γ1-γ4. The π and κ clades were named after P and K88 pili, commonly found in *E. coli* isolates, respectively, whereas both α and σ clades represent the nonclassical CUP [26]. Fimbrial gene expression, biogenesis, structure, surface sensing and prevalence of *E. coli* have been nicely reviewed elsewhere [16,19,38,39,40,41]. The nonclassical CUP are further differentiated into alternative (or alternate) and archaic CUP. Despite being composed by chaperone and usher proteins, the alternative and archaic pathways differ significantly from the classical one, both in terms of the amino acid sequences of the proteins involved and in their assembly mechanisms. 

The alternative CUP is mainly represented by class 5 pili found in β- and γ-proteobacteria; the most known example includes eight members that are produced by enterotoxigenic *E. coli* (ETEC). However, several pili of this clade were found in *B. cepacia*, *Salmonella enterica*, *Yersinia pestis*, *Aeromonas hydrophila*, and *Shewanella oneidensis* [26]. Vice versa, the archaic pathway includes the most widespread pili among bacteria, being found in all Proteobacteria, in Cyanobacteria and *Deinococcus-Thermus* phyla [26]. The archaic CUP includes the CsuC−CsuD pathway found in *A. baumannii* (Figure 1). The strong biofilm-forming activity that characterizes this bacterium allows its long-lasting persistence on abiotic surfaces and colonization of different medical devices [42]. Genetically, Csu pili are organized in a polycistronic operon, *csuA*/*BABCDE*; the major pilin subunit, CsuA/B dimer, adaptor monomers CsuA and CsuB and the tip adhesion, CsuE, are assisted and trafficked to the outer membrane by the CsuC chaperone and the CsuD usher proteins [43]. Expression of the CsuC−CsuD pathway is controlled by a two-component system composed by the sensor kinase BfmS and the response regulator BfmR [44,45,46]. Alterations in the expression of *bfmR* such as deletion or overexpression significantly affect biofilm formation [44]. A second two-component system controls, at least partially, the expression of these pili, the GacSA [47]. In addition to the *csu* operon, the GacSA system acts as a global virulence regulator that controls the expression of 674 genes involved in virulence, biofilm, motility, metabolism of aromatic compounds, citrate, and resistance against human serum [48]. It is interesting to note that, differently from the other two-component systems, *gacS* and *gacA* genes might be not genetically linked but encoded in distinct parts of the genome, which might vary among different *A. baumannii* strains [49]. 

### 2.2. Role of Afimbrial Structures

In Gram-negative bacteria, the OM represents the outermost structure of the bacterial cell that acts as a physical and mechanical barrier for the maintenance of basic cellular physiology against detrimental molecules such as antibiotics. Several integral membrane proteins arranged in β-barrels are embedded within this membrane; these outer membrane proteins (OMPs) consist of 8 to 26 strands that cross the OM with large external and short internal loops. OMPs can be monomeric or trimeric and classified as specific transporters (e.g., the maltoporin LamB) and nonspecific channels (e.g., the porin OmpC), both involved in the uptake of small molecules fundamental for cell physiology [50]. Genetically related OmpA porins are highly abundant OMPs among Gram-negative bacteria, being involved in a variety of pathogenic roles, including adhesion, invasion, biofilm, serum resistance, evasion of host defenses and antimicrobial resistance [51,52,53,54]. In *E. coli* and *A. baumannii*, OmpA increased biofilm formation on several plastic surfaces (e.g., polystyrene, polypropylene, and polyvinyl chloride) (Figure 1) [55,56]. The gene *ompA* is commonly found among strong biofilm-forming *A. baumannii* clinical strains together with *csuE*, *bfmS*, *pgaB*, *ptk*, and *epsA* genes. The *pgaB* gene is located within the *pgaABCD* operon and it is involved in the synthesis of poly-β-(1-6)-N-acetylglucosamine (PNAG), whereas *ptk* and *epsA* genes encode proteins involved in the synthesis of the capsular polysaccharide and the exopolysaccharide exporter, respectively, both critical for biofilm formation, bacterial protection and aggregation [57]. It was recently found that PNAG, the major component of the *A. baumannii* biofilm matrix, has a surface presentation and accessibility to bacterial external proteins (also known as lectins) remarkably different from Gram-positive bacteria such as *S. aureus* (Figure 1) [58]. These PNAG characteristics not only contribute to biofilm integrity, but also confer to *A. baumannii* an extraordinary tolerance to desiccation stress, and improved persistence in natural and healthcare environments [58]. As mentioned before, the self-regulated BfmS/BfmR system plays a central role in cell envelope structures in *A. baumannii* (Figure 1) [59]. Apart from the *csu* operon encoding the chaperone/usher pili, BfmR regulates the K locus genes for exopolysaccharide production, which is crucial for biofilm formation on abiotic surfaces [59,60]. Interestingly, *bfmR* mutants lost the biofilm-forming activity, resistance to some antibiotics, and showed aberrant cell morphologies; in contrast *bfmS* mutants acquired a hypervirulent phenotype for exopolysaccharide overproduction [59,60]. In addition to these proteins, Bap was shown to play crucial roles in *A. baumannii* biofilms; this 854 kDa protein is composed of 8620 amino acids arranged in tandem repetitive modules, from A to G (Figure 1) [61]. This large surface protein is involved in water channel formation and biofilm formation by *A. baumannii*, both on abiotic and biotic surfaces. Despite common features, the *bap* gene showed size and nucleotide variations among the *A. baumannii* isolates [61]. Nevertheless, the general three-dimensional structure of the immunoglobulin-fold domain is conserved among Baps that self-assemble to form amyloidogenic oligomers [62]. The inability of a *bap* mutant to support the biofilm mature architecture demonstrated the essential role of Bap in *A. baumannii* biofilms [63]. Other proteins involved in *A. baumannii* biofilms are FhaB/C proteins as well as the trimeric Ata autotransporter, belonging to the type V secretion systems; these systems are composed by a transporter embedded within the OM that facilitates the export of exoproteins (Figure 1) [64]. As for other Gram-negative bacteria, these systems are involved in the interactions of the bacteria with their environment [64,65]. In fact, these adhesins were shown to play a role in adherence and biofilm formation on both abiotic and biotic surfaces, therefore increasing the colonization potential of *A. baumannii* [22,65,66,67]. 

Biofilm formation is fine-tuned by QS, an intra- and interspecies density-dependent gene regulation that coordinates several important biological processes in bacteria. QS depends on the release and sensing of QS-signaling molecules or autoinducers in the extracellular milieu. There are at least two major types of QS: type I is specific for intraspecies communication, whereas type II is involved in interspecies communication, allowing bacteria to respond to their own and other species autoinducers. Type I QS produces N-acyl homoserine lactone (AHL) or autoinducer-1, whereas type II QS signals via autoinducer-2. The QS system is based on a synthase responsible for producing QS molecules that modulate a transcription factor, which in turn, tunes the expression of several genes, including those involved in QS systems. Therefore, QS has a key role in bacterial gene regulation. For a more comprehensive picture of the complex impact of QS in modulating biofilm formation in *A. baumannii*, please refer to [68]. 

### 2.3. Role of Outer Membrane Vesicles (OMVs)

OMVs are small, round-shaped bilayer vesicles that are released into the extracellular milieu by several Gram-negative bacteria and contain proteins, lipids, nucleic acids, and other bacterial metabolites (Figure 1) [69,70,71]. Although secreted during normal bacterial growth, environmentally stressful conditions (e.g., temperature, nutrient deficiency, antibiotics) increase OMV production [69]. Under these conditions, OMVs are needed to establish a colonization niche that enable bacteria to create the most favorable conditions for biofilm formation, and/or survival in the host [69]. Several models have been proposed for OMV biogenesis in Gram-negative bacteria; it was suggested that OMVs bleb from the OM via a specific and local reduction in the cross-linking between peptidoglycan and OMPs [69,72]. Their surface is composed of lipopolysaccharide, phospholipids and OMPs, while the vesicular lumen contains periplasmic and cytosolic proteins, nucleic acids, metabolites and signaling molecules [69,72]. OMVs are relevant components of biofilms with specific contents tailored to this bacterial lifestyle [69,72]. 

*A. baumannii* OMVs enclose several virulence factors, including proteins involved in biofilm formation, host cell adhesion, motility, and antibiotic resistance [73]. One of the most abundant OMPs within the OMVs is OmpA, a major virulence protein in *A. baumannii* [69]. A recent study demonstrated that BfmS controls the amount of OmpA within the OM; lowering its OM content increases OMV production as a consequence of the reduced interactions of the C-terminal domain of OmpA with diaminopimelate of the peptidoglycan layer [74]. A clear link between biofilm-forming activity and OMVs was shown in *A. baumannii* isolates during the reversible switch from opaque to translucent [75]. The more piliated translucent variants showed a significant increase in biofilm-forming activity as well as OMV production [75]. Therefore, it is believed that OMVs deliver components of the EPS to fasten biofilm maturation and enhance bacterial survival.

In addition, it became evident that bacteria also use cytoplasmic proteins outside the cytoplasm, known as moonlighting proteins. In general, these proteins are involved in bacterial metabolism, protein synthesis and folding, and DNA replication. Different mechanisms of secretion have been proposed, such as a direct release, embedding on OMVs or binding to the cell envelope [76]. Interestingly, the outside activity of these moonlighting proteins is totally different from their inside-cytoplasmic activity. In *A. baumannii*, the translation elongation factor Tuf was shown to be a moonlighting protein; this protein participates in serum resistance and tissue dissemination (Figure 1) [77]. Based on this evidence, it can be speculated that other adhesive moonlighting proteins could contribute to biofilm formation in *A. baumannii*, as reported for other Gram-negative bacteria [78]. 

## 3. Are MDR Bacteria More Able to Form Biofilm?

Historically, bacterial persistence is the complex phenomenon leading to resistance to antibiotic killing activity without expressing any specific resistance mechanism [79]. However, it was observed that in rapidly growing cultures some cells can switch into slowly replicating ‘‘persister’’ subpopulations that could be maintained for several generations. Hence, the selection of persistent bacteria is a natural phenomenon aimed at increasing the species survival in fluctuating environments [79]. 

Despite several hypotheses and experimental data trying to unveil bacterial persistence, this phenomenon is not completely characterized yet. Dormancy, latency, and persistence are terms used to describe the capacity of pathogenic bacteria to arrest their growth in response to host- or environmental-imposed stresses. Slow to negligible replication is commonly observed in microbes because they often inhabit harsh environments and face environmental stresses that are incompatible with rapid growth [80]. Undoubtedly, one of the most studied mechanisms leading to bacterial persistence is the formation of biofilm. The most critical aspects of biofilms formed by pathogenic bacteria are: (i) synthesis of the EPS for biofilm structure and mechanical stability; (ii) retention of extracellular enzymes for nutrient acquisition and for antimicrobial detoxification; (iii) enhanced tolerance to disinfectants, biocides, and other stressors; (iv) enhanced intercellular communication for a coordinated virulence gene expression and for the activation of horizontal gene transfer systems.

### 3.1. The Success of A. baumannii Biofilms

Regretfully, *A. baumannii* is widely recognized as a healthcare-associated pathogen. Two aspects contribute to its success: its capability to grow and persist as a biofilm within hospitals and healthcare facilities, as well as within a host, leading to chronic/persistent infections. Among Gram-negative bacteria, after *P. aeruginosa*, *A. baumannii* is prevalent in causing HAIs, as well as in contaminating hospital environments [81,82]. *A. baumannii* is well known for possessing several antibiotic resistance determinants; however, the capability to form biofilm poses another relevant issue in fighting these MDR bacteria.

It was reported that *A. baumannii* can form biofilm in hospital settings by modulating the expression of different proteins, thereby leading to increased levels of resistance to antimicrobials, including antibiotics and detergents, and tolerance to desiccation [83]. Moreover, this expression pattern could also induce the development of the cell dormancy state by reducing the metabolic activity and increasing the cell adherence through a total restyling of both the OM and the OMPs. The OM represents one of the most important components mediating biofilm formation; hence, a deep characterization of its composition could help in defining critical changes promoting biofilm-forming activity. The *adeRS* genes encode for the two-component system regulating the expression of several OM efflux pumps in *A. baumannii*. In fact, mutations in *adeRS*, resulting in the overexpression of Ade proteins including AdeB, are commonly responsible for the increase of *A. baumannii* antimicrobial resistance. Accordingly, it was reported that deletion of *adeRS* and *adeB* genes resulted in increased antibiotic susceptibility and biofilm impairment on both biotic and abiotic surfaces. Hence, it can be concluded that the expression of efflux pumps is positively correlated with the capability to form biofilm in *A. baumannii* [84]. Similarly, overexpression of efflux pumps has been recognized as a common strategy to improve biofilm-forming activity both in *E. coli* and *P. aeruginosa*, by regulating the efflux of EPS and QS molecules and promoting cell aggregation [85].

Lipid A modification is directly involved in antimicrobial resistance and in biofilm formation. It has been shown that acylation of lipid A leads to the resistance to cationic peptide drugs, such as colistin, and increases the desiccation tolerance, thereby possibly facilitating the colonization of clinical settings (Figure 1) [86]. In addition, the enhanced survival rates of bacteria in an in vivo model of catheter infection were due to high levels of lipid A palmitoylation in biofilm-forming bacteria [87]. This mechanism could contribute to biofilm tolerance to host immune defenses as well as to the development of chronic infections.

To assess the impact of lipid A modifications on biofilm formation in *A. baumannii*, Farshadzadeh et al. compared colistin-susceptible isolates and their laboratory-selected colistin-resistant counterparts for biofilm-forming ability, using the microplate biofilm assay and in vitro and in vivo catheter-adhesion models. Lipid A modifications did not alter the capability to produce biofilm and both susceptible and resistant isolates produced biofilms to the same extent. Moreover, no differences were observed in the expression of biofilm-associated genes between the two sets of isolates [88]. It was also reported that the modification of lipid A is extremely costly for bacterial cells and for this reason, several papers described a reduced bacterial growth rate and biofilm formation in colistin- and polymyxin B-resistant *A. baumannii* strains [89]. Hence, the mechanisms through which modifications of lipid A may contribute to biofilm formation are far from fully understood.

Contrarily, it was reported that exposure to subinhibitory levels of some antibiotics, below the minimal inhibitory concentrations (sub-MICs), promote biofilm formation in *A. baumannii* clinical isolates. In particular, sub-MICs of colistin, levofloxacin and meropenem induced the overexpression of both the AdeFGH efflux pump and the autoinducer synthase AbaI, which are positively associated with increased biofilm-forming activity [90,91]. Like members of the LuxI family of autoinducer synthases, AbaI is responsible for normal biofilm development in *A. baumannii* [92]. Moreover, the overexpression of Bap and PNAG in cells exposed to sub-MICs of colistin and polymyxin B was observed, underlining a biofilm-mediated response to the presence of these antibiotics [91]. It is noteworthy that this phenotypic change could represent an important threat if bactericidal concentrations are not reached at the infection site.

Sung-Ho Yun et al. provided extensive data on the proteomic content of the MDR *A. baumannii* strain DU202 cultivated in the presence of tetracycline and imipenem [93]. They showed that both AdeABC and AdeIJK were induced in the presence of tetracycline and, to a lesser extent, of imipenem (Figure 1). Moreover, another 11 resistance nodulation-division (RND) transporters (A1S_0255, 0535, 0537, 0538, 0908, 1242, 1243, 2618, 2619, 2620, and 2736) as well as outer membrane lipoproteins (A1S_2613 and 2611) were upregulated [93]. An increased expression of proteins involved in the oxidative stress response in *A. baumannii* AB5075 was shown upon imipenem exposure [94]. Since most of these proteins are in the OM, it can be concluded that antibiotic exposure induces major OMP changes, thereby potentially influencing bacterial adhesion. Accordingly, tetracycline and imipenem induced the expression of BfmS, known to activate the reshaping of the bacterial cell wall upon entry into the post-exponential phase of growth. In addition, it was reported that BfmRS increased tolerance to bactericidal antibiotic exposures, enhanced the expression of genes mainly involved in the protection against oxidative and osmotic stresses and regulated biofilm formation during the post-exponential growth phase or during starvation [44,59]. 

Overall, it can be concluded that biofilm formation is directly linked to exposure to sub-MICs of different antibiotics, creating a positive feedback that allows the switch from planktonic to sessile growth. It is worth mentioning that MDR *A. baumannii* strains are more prone to form biofilm due to their intrinsic OM characteristics, such as the different OMP profiles and the presence of multiple efflux pumps. On the other hand, biofilms enhance their desiccation tolerance. Accordingly, Greene et al. found that *A. baumannii* clinical MDR strains were more resistant to desiccation than the non-MDR strains [95]. Conversely, environmental isolates collected from a hospital setup were higher biofilm producers compared to clinical isolates associated with patients. Therefore, desiccation tolerance and biofilm producing activity cooperate to increase bacterial survival, although biofilms play a major role in hospital environmental isolates [95]. The different extent of desiccation tolerance could be related to contrasting habitat adaptation between clinical and environmental bacteria. It is believed that the acquisition of the MDR phenotype among environmental isolates imposes a decrease in desiccation tolerance, which must be counterbalanced by an increase in the ability to form biofilms for bacterial survival. Accordingly, healthcare environmental bacteria retain a certain level of antibiotic susceptibility to preserve desiccation tolerance in association with their biofilm positive phenotype. Accordingly, Rodriguez-Baño et al., showed that biofilm-forming *A. baumannii* isolates were more susceptible to imipenem and ciprofloxacin compared with their non-biofilm-forming counterparts, which indicates that the survival of these isolates in the hospital environment is less dependent on antibiotic resistance with respect to biofilm formation [96]. The bacterial lifestyle “inside and outside” the human host and in the clinical setting, respectively, seems to induce different gene expression profiles; during host colonization, bacteria maximize the expression of antibiotic resistance mechanisms; in contrast, during healthcare environmental colonization, bacteria enhance the expression of long-term survival mechanisms [28,97]. These different lifestyles impose a fitness cost; the expression of antibiotic resistance genes is often manifested as reduced growth rates, while the expression of long-term survival genes implies antibiotic susceptibility. Nevertheless, due to the high genotypic and phenotypic diversity of *A. baumannii* strains colonizing the hospitals and communities, we cannot exclude a continuous habitat-driven switch between environmental and clinical isolates. 

Differently, Aziz et al. observed that *A. baumannii* isolates carrying the extended-spectrum β-lactamase *blaPER-1* gene formed a significantly greater amount of biofilm than isolates that were *blaPER-1*-deficient [98]. Interestingly, BlaPER-1 was shown to increase the adherence to epithelial cells, thus suggesting its involvement in the early stages of biofilm formation (Figure 1). Thummeepak et al., screened 225 clinical *A. baumannii* strains for biofilm formation and antibiotic resistance gene expression [99]. No correlation was found between MDR or extensively resistant (XDR) phenotypes and the extent of biofilm formation; however, when a single drug resistance was considered, they found that gentamicin resistant isolates were the highest biofilm producers. They concluded that the impact of antibiotic resistance on biofilm formation is strain- and antibiotic-dependent [99]. Aliramezani et al. surveyed *A. baumannii* isolates collected from hospital furniture (bedrail, bed sheet, dialysis machine, etc.,) as well as from medical devices (thermometer, stethoscope, blood pressure monitoring, etc.,) and found that both carbapenem-susceptible and -resistant strains were able to form biofilm without any significant differences between the two groups [100]. In addition, by analyzing 64 *A. baumannii* isolates from burn infections, Amin et al. observed that all strong biofilm-former strains were non-MDR, whereas those classified as weak biofilm-formers were MDR [101]. In contrast, 154 strong biofilm-producer *A. baumannii* clinical isolates were found resistant to several antibiotics, including ticarcillin, ceftazidime, gentamicin, and piperacillin [102]. Among these 154 strains, the strongest biofilm producers were resistant to piperacillin, thereby highlighting a positive correlation between the resistance to penicillins and biofilm formation [102]. In addition, carbapenem-resistant isolates collected from bloodstream infections were shown to produce a thicker and more uniform biofilm with respect to those susceptible, confirming this positive correlation [103]. These contrasting data could be partially justified by different expression levels of OmpA, which is known to be involved in both biofilm formation and resistance to β-lactams. Therefore, high levels of OmpA could contribute to strong biofilms and increased antibiotic susceptibility and vice versa. Although the general mechanism by which biofilm contributes to antibiotic resistance is known, there is a profound gap in explaining the impact of OM changes observed in MDR and XDR *A. baumannii* on the capability to form biofilm. Whether MDR and XDR bacteria are more able to form biofilm is still an open question. Gaining this knowledge could clarify the extent of environmental contamination by resistant *A. baumannii* strains and, therefore, the targeted strategies to adopt with the aim of lowering HAIs.

### 3.2. Biofilms as a Source of Bacterial Dissemination

In mature biofilms, cell shedding is commonly observed: the dispersion of the so called “streamer” cells aimed at bacterial dissemination and colonization of new surfaces [104,105,106,107]. However, it is still unclear whether biofilms differentiate into subpopulations before or during the dispersion phase, and/or a transition stage exists between biofilm and the planktonic lifestyle [107,108]. Both unicellular and multicellular life phases alternate over time [107,108,109]. Interestingly, it is supposed that dispersed bacterial cells are heterogeneous, possessing diverse phenotypes compared to biofilm-embedded and planktonic cells. Although detailed investigations of the phenotypic switch during biofilm formation are limited, several studies have shown that planktonic cells and biofilm communities are associated with a unique transcriptional behavior of distinct sets of genes [108,109,110]. The Clusters of Orthologous Groups analysis showed that the most representative gene functions in streamers cells were associated to translation, ribosomal structure, biogenesis, amino acid transport and metabolism. These findings suggest that biofilm-dispersed cells are metabolically more active and possess distinct metabolic signatures compared to both biofilm and planktonic cells [107,108,109]. Further studies are needed to evaluate if these specific gene expression profiles could be used as potential biomarkers of the different bacterial lifestyles.

## 4. Clinical Impact of *A. baumannii* Biofilms

As outlined before, the ability of *A. baumannii* to grow as a biofilm represents an important virulence factor. Indeed, *A. baumannii* can retain or even increase its capacity to form biofilms if rehydrated after 60 days desiccation on a solid surface, despite a considerable reduction in culturability over time [111]. Rapid adaptation was also observed, both to temperature shift (from room temperature to 37 °C), and the availability of nutrients (from starvation to food availability), conditions that bacteria can easily find in a new patient [111]. In other studies, biofilms formed by MDR *A. baumannii* presented significantly higher resistance to desiccation and to common disinfectants, such as ethanol, benzalkonium chloride and chlorhexidine, than their planktonic counterpart [112,113]. In this way, *A. baumannii* might persist in clinical devices and hospital environments, thus causing HAIs and outbreaks worldwide [95,114]. Supporting this, several studies have reported biofilm formation ability in *A. baumannii* strains causing various types of infection: surgical site and wound infections [115,116], urinary tract infections [117,118], blood and respiratory infections, and skin and soft infections [119,120,121].

It is worthy of note that *A. baumannii* strains causing infections displayed a significantly higher biofilm production potential compared with those originating from colonization sites, thus indicating that biofilm formation is critical in establishing an infection in *A. baumannii* [120]. Particularly, in the case of tracheal tubes, the device-related strains were significantly more efficient in forming biofilm than non-device-related strains [120]. In another study, the biofilm-producing isolates were found to colonize the respiratory tract for longer times compared with those not able to form biofilm, as clearly indicated by the median duration of colonization (18 vs. 12 days, respectively; *p* < 0.05) [122]. Furthermore, during colonization, *A. baumannii* biofilm producers showed a marked social behavior by promoting co-colonization with other bacterial species, particularly *S. aureus*, to form polymicrobial biofilms [122]. The physical interaction of *A. baumannii* with other bacterial species might indeed result in increased fitness due to cooperative metabolism and community development, and a synergistic potential for increased pathogenicity, as already observed in *A. baumannii−Porphyromonas gingivalis* oral mixed biofilms [123].

Biofilm formation seems not to be relevant in the *A. baumannii* epidemic spread. In this regard, Hu et al. found that sporadic isolates have significantly greater biofilm-forming capabilities than the outbreak and epidemic isolates [124]. Other studies indicated that biofilm formation is generally a clone-specific feature [125,126]. For example, a stronger ability to form biofilm was observed among clinical isolates of *A. baumannii* belonging to the European clone II [126]. Similarly, Sanchez et al. found that isolates belonging to selected pulsotypes were associated with a greater ability to form biofilms [127].

### Device and Non-Device Related Biofilm Infections

In vivo biofilm formation by *A. baumannii* has also been documented. Talreja et al. found that ocular isolates forming higher amounts of biofilm caused higher retinal damage, thus suggesting that biofilm formation is involved in the pathogenesis of *A. baumannii* endophthalmitis [128]. In a murine wound model, *A. baumannii* AB5075 MDR strain formed robust biofilms within and above the wound bed on the occlusive dressing [129]. It is also worth noting that this strain could not form a relevant biofilm amount in vitro, thus highlighting there are cues in the in vivo environment that trigger biofilm assembly [129].

Several reports have also indicated the presence of *A. baumannii* biofilm on different hospital materials, such as latex, anodized aluminum, stainless steel, and polycarbonate surfaces [47,55,95,130]. However, the interaction of different clinical strains with abiotic surfaces has been reported to vary according to the specific features of both surface and strain [131]. Particularly, Greene et al. showed that polycarbonate—a low-cost, durable plastic preferred for its ability to undergo autoclaving—is the most accommodating surface for biofilm growth, making it an ideal reservoir [95]. In another study, Lin et al. showed that rubber latex was the tubing material most susceptible to *A. baumannii* biofilm formation, followed by polyvinyl chloride and silicone [132].

Other in vitro and in vivo studies have documented that *A. baumannii* shows relevant propensity to grow as a biofilm on several abiotic surfaces, causing the surface colonization of hospital equipment and indwelling medical devices, such as catheters, endotracheal tubes, and dialysis tube bottles [96,120,130,133,134]. These findings might explain why the increased use of prosthetic interventions—e.g., mechanical ventilation and central venous and urinary catheterization—have greatly increased the incidence of *A. baumannii* infections [85,135,136]. In this regard, Rodriguez-Bano et al. found that biofilm can be relevant to the pathogenesis of some device-associated *A. baumannii* infections, such as those involving Foley and venous catheters, as well as cerebrospinal fluid shunts [96]. Particularly, *A. baumannii* forms polymicrobial biofilms on the inner surfaces of urinary catheters, consisting of a dense and interconnected network of bacteria with different shapes and sizes, surrounded by a relevant amount of exopolysaccharide matrix [130]. In agreement, several authors reported the presence of *A. baumannii* in biofilms formed in the endotracheal tubes of ventilated patients, with a higher predominance in adults [133] compared with pediatrics [134]. Finally, systemic infections caused by carbapenem-resistant *A. baumannii* were unlinked to biofilm formation on extracorporeal membrane oxygenation catheters [137].

## 5. Prevention and Treatment Strategies against *A. baumannii* Biofilm

The treatment of biofilm-associated *A. baumannii* infections represents a great challenge as this microorganism is responsible for chronic infections. As mentioned before, the ability to form biofilm enhances *A. baumannii* resistance to antibacterial drugs and allows it to circumvent the host immune-mediated clearance. As our current arsenal of antimicrobials has become increasingly less effective at treating these infections, there is a serious demand for alternative therapeutic approaches as valid options to the often unsuccessful classical antibiotic treatment therapies.

### 5.1. Inhibition of Biofilm Formation 

The most preferable strategy may be to attempt to prevent biofilm formation from occurring in vivo. Since the adherence to biotic or abiotic surfaces is a preliminary step in the formation of bacterial biofilm, disrupting or limiting bacterial adhesion would avoid biofilm-related antibiotic resistance and aid the immune system in clearing the infection. The potential targets for therapeutics recently described to prevent biofilm formation by *A. baumannii* are summarized in Table 1.

#### 5.1.1. Modulation of Genes Involved in Biofilm Formation

Due to its key role in *A. baumannii* biofilm formation, the Bap protein may represent a potential therapeutic target. Considering that medical implants—e.g., pacemakers, catheters, and mechanical heart valves—are made of hydrophobic materials (teflon, stainless steel, silicon, etc.), hydrophobic microorganisms are adhering to them relatively easily [166]. As already outlined, *A. baumannii* binding to abiotic sites is achieved through the Csu pili as well as OmpA (Figure 1) [43,55,167]. 

Several studies have reported the antiadhesive properties of bioactive compounds, such as phenolics, essential oils, terpenoids, lectins, alkaloids, flavonoids, and polypeptides [168]. Abirami et al. showed that the concentration-based antibiofilm effects of pyrogallol, a polyphenolic organic compound found in the galls and barks of several trees, was due to reduced swarming motility, and the downregulation of genes involved in cell adhesion (*ompA*, *csuA/B*), and biofilm formation and stabilization (*bap*) [138]. Similarly, exposure to myrtenol, a bicyclic monoterpene present in various plants, caused a strong reduction in biofilm thickness and surface coverage secondary to reduced cell surface hydrophobicity, motility (swarming and twitching), and expression of the biofilm-associated genes such as *bfmR*, *csuA/B*, *bap*, *ompA*, *pgaA*, and *pgaC* [139]. In another study, Raorane et al. screened 12 flavonoids for *A. baumannii* biofilm inhibition [140]. Curcumin was the most effective against several clinical, MDR isolates of *A. baumannii* causing a significant and dose-dependent reduction of biofilm formation by 46 and 93% at 20 and 100 mg/mL, respectively. In vitro and in silico findings indicated that the binding efficacy of the flavonoid to the two-component response regulator BfmR, which acts as a master control switch for biofilm development [141], was correlated with antibiofilm efficacy [140].

In a recent study, it has been shown that the antibiofilm activity of 5-hydroxymethylfurfural, an organic compound formed by the dehydration of reducing sugars, is dependent on the downregulation of biofilm-related *bap*, *csuA/B*, *ompA*, *bfmR* and *katE* genes, which in turn affects *A. baumannii* biofilm formation and maturation, cell adherence, cell surface hydrophobicity and EPS production [142]. Similarly, the antimicrobial peptide cecropin Cec4 inhibits the expression of several genes involved in biofilm formation, namely the pilus-related gene *csuE*, the two-component regulatory system genes *bfmR* and *bfmS*, and the biofilm-related *bap* gene [143].

Probiotics have attracted attention as safe and beneficial therapeutics [169]. A study by Shin and Eom reinforced the value of *Clostridium butyricum* as a probiotic and suggested its potential as a new therapeutic alternative against *A. baumannii* [144]. Indeed, they found that cell-free supernatants derived from *C. butyricum* affected biofilm formation by *A. baumannii*, probably due to the inhibition of motility and of expression of RND-type efflux pump-related *adeABC* genes.

Several polypeptides targeting OmpA have been proposed to prevent *A. baumannii* adhesion to surfaces. AOA-2, a cyclic hexapeptide, acts as a blocking agent of OmpA, thus decreasing the adhesion of *A. baumannii*, *P. aeruginosa* and *E. coli* to both biotic and abiotic surfaces; furthermore, it significantly enhances the sensitivity of *A. baumannii* to colistin [170]. In vivo findings confirmed the protective role of AOA-2 (10 mg/kg) in combination with colistin (10 mg/kg) in a mouse model of sepsis [170,171]. In addition, some classic antimicrobial peptides (AMPs) interacting with OmpA have been gradually discovered. For example, LL-37 interacted with the amino acid residues 74–84 of *A. baumannii* OmpA in a dose-dependent way, decreasing the bacterial adhesion to host cells [172]. These polypeptides, specifically targeting *A. baumannii* OmpA without bactericidal activity, may avoid triggering bacterial evolution pressure and could be used alone or synergistically combined with other antibacterial compounds.

#### 5.1.2. Inhibition of QS Signals

As mentioned above, in *A. baumannii*, biofilm formation is dependent on the activation of a typical LuxI/LuxR-type QS network involving *abaI* synthase, *abaR* receptor and various AHLs not homogenously distributed among species, although 3-hydroxy-dodecanoyl-L-homoserine lactone (3-OH-C12-HSL) is the prevalent one [68,173]. Recently, inhibitors of the QS process have been developed as a possible strategy for the design of new agents able to reduce *A. baumannii* biofilm formation without interfering with the bacterial growth. QS inhibitors (QSIs) induce synthase or receptor inactivation via competitive binding, whereas quorum quenching (QQ) enzymes switch off signal transduction through the enzymatic degradation of signal molecules [174].

The inhibition of QS signals has been frequently proposed as a new therapeutic approach in recent years. Saroj and Rather found that streptomycin at sub-MIC acted as an antagonist of 3-OH-C12-HSL, thus interfering with the signal binding to the AbaR protein [175]. In another study, the screening of a library focused on synthetic, non-native AHLs showed that the non-natural ligands containing aromatic acyl groups were the most effective AbaR antagonists, thus inhibiting the formation of biofilm in *A. baumannii* [145]. Other studies gave evidence for the potential of some biological extracts or natural products to inhibit biofilm formation and QS. Siphonocholin (Syph-1), a marine steroid isolated from *Siphonochalina siphonella*, at sub-MICs inhibits the biofilm and pellicle formation in *A. baumannii*, probably due to anti-QS activity, and reduces EPS production and swarming motility [146]. In addition, the activity-guided by the partially purified fraction F1 derived from *Glycyrrhiza glabra* leads to a significant reduction in QS-mediated virulence of *A. baumannii* by downregulating the expression of the autoinducer synthase gene, *abaI*, and consequently reducing the production of 3-OH-C12-HSL [147]. Furthermore, Alves et al., found that linalool, the major oil compound from *Coriandrum sativum*, inhibited the formation of biofilm and dispersed established biofilms by *A. baumannii* on several clinically relevant surfaces, both by affecting bacterial adhesion and interfering with the QS system [148]. Confirming the antibiofilm potential of the essential oils, in another study Ismail et al. observed that the essential oil from the leaf of the myrtaceous plant *Pimenta dioica* at 0.05 µg/mL could efficiently inhibit and eradicate *A. baumannii* biofilm by 85 and 34%, respectively [176].

Pentacyclic triterpenoids have been implemented in antibiofilm research recently. In this regard, betulinic acid, glycyrrhetinic acid, and ursolic acid affected biofilm formation and structure in *A. baumannii* by strongly interacting with major QS regulators, AbaI and AbaR, in conserved pockets [149].

Photodynamic therapy has been shown to be able to reduce the expression profile of the QS system genes associated with biofilm formation by *A. baumannii*. Pourhajibagher et al. observed that in *A. baumannii*, *P. aeruginosa*, and *S. aureus* multispecies biofilms photosensitized with indocyanine green, the expression levels of *abaI*, *agrA*, and *lasI*, were downregulated, respectively [177]. These findings may have potential implications of photodynamic therapy for the treatment of multispecies bacterial biofilms involved in burn wound infections.

QQ can be achieved by QQ enzymes, AHL lactonase, that hydrolyze the quorum signal molecules. In *A. baumannii*, the signal molecule AHLs bind to receptor molecules on the cell surface and initiate the QS process. Targeting AHL synthase may be, therefore, an effective QQ strategy to interrupt QS and affect biofilm formation [178]. It has been shown that MomL, a newly discovered QQ enzyme, can effectively degrade different AHLs of various Gram-negative bacteria. In particular, it reduced biofilm formation and increased biofilm susceptibility to different antibiotics in *A. baumannii* [119]. QQ strategy could be also combined with other enzyme treatment methods as an alternative approach to prevent the colonization and survival of the pathogen on the surface. In this regard, Mayer et al. observed that the combined action of QQ enzyme lactonase Aii20J and DNase could significantly reduce the biofilm formation of *A. baumannii* ATCC 17978 [150]. For the first time, Chow et al. demonstrated the use of recombinant QQ enzymes in the disruption of biofilm formation (i.e., reduced biomass and thickness) by *A. baumannii* [179].

Targeting AbaR in its receptor or regulatory activities could eventually contribute to the invalid binding of AHLs, thus quenching the QS system. Recently it has been observed the antibiofilm activity of monounsaturated chain fatty acids, palmitoleic acid, and myristic acid against *A. baumannii* ATCC 17978. These fatty acids caused a biofilm dispersing effect and drastically reduced motility by decreasing the expression of the regulator from the LuxIR-type QS communication system AbaIR, thereby reducing the production of AHL [151].

Altogether, these findings highlight that there are great opportunities to deal with the problem of *A. baumannii* biofilm formation by affecting the QS system. Furthermore, efficacy and toxicity studies, under the condition of simulating “real” infection, enrolling clinical rather than laboratory strains are, therefore, warranted to determine the current potential of QSIs or QQ enzymes as therapeutic antibiofilm drugs. In addition, these antibiofilm approaches could be used in synergistic combination with other antimicrobials or with each other, e.g., to functionalize the surface of catheters or implants to prevent biofilm formation. Finally, the standardization of methods for screening new QSI candidates and exploring their specificity remains to be explored.

#### 5.1.3. Inhibition in EPS Production

EPS is an important component of bacterial biofilms and helps to maintain a structured multicellular bacterial community. In *A. baumannii*, EPS is primarily composed of carbohydrates, with mannose constituting a major part of the extracellular polysaccharides [180]. Bioactive compounds, such as pyrogallol and myrtenol, were shown to be active against both biofilm formation and mature *A. baumannii* biofilms causing a strong reduction in biofilm thickness and surface coverage secondary to reduced EPS production [138,139]. Triterpenoids representing the three major families of triterpenoids (betulinic acid, glycyrrhetinic acid, and ursolic acid) were demonstrated to severely attenuate EPS production, affecting the overall structure of biofilms, as depicted by scanning electron microscopy (SEM) analysis [149]. In another work, exposure to synthetic QSIs n-C10H21 or 3-ClBn (3-hydroxy-2,3-dihydroquinazolin-4(1H)-one analogues) caused a significant reduction of the extracellular polysaccharides of *A. baumannii* [181]. Exposure to Al_2_O_3_ nanoparticles (NPs) caused a significant loss of protein and carbohydrates in EPS that damaged the biofilm architecture [152]. 

Another interesting strategy might consist in inhibiting EPS export. In *A. baumannii*, EPS synthesis and export are regulated by the bacterial Wza-Wzb-Wzc system [182]. The protein tyrosine phosphatase Wzb mediates dephosphorylation of Wzc that is required for the export of the EPS through porin Wza-Wzc complex (Figure 1). Tiwari et al. found that labetalol hydrochloride, an effective agent in essential hypertension, may inhibit the interaction of Wzb with Wzc, block the dephosphorylation of Wzc, hence inhibiting EPS production and export that eventually affects bacterial motility, pathogenicity, and biofilm-forming abilities [153].

#### 5.1.4. Inhibition of Efflux Pumps

Several studies have demonstrated the key role played by efflux pumps in bacterial biofilm formation, particularly in ESKAPEE pathogens [183]. The expression of such pumps is indeed shown to be upregulated in biofilms, thus leading to increased antibiotic resistance. Consequently, efflux pump inhibitors (EPIs) might have the potential for antibiofilm agents. In this frame, phenylalanine-arginine β-naphthylamide (PAβN), 1-(1-naphthylmethyl)-piperazine (NMP) and carbonyl cyanide 3-chlorophenylhydrazone (CCCP) are among the most studied synthetic inhibitors in *A. baumannii* [183]: PaβN is active against the AdeFGH pump by reducing the MIC of trimethoprim, chloramphenicol and clindamycin (AdeFGH substrates); the aryl-piperazine derivative, NMP enables a significant reduction in the MIC of fluoroquinolones and aminoglycosides (AdeABC substrates); CCCP is a synthetic inhibitor of proton motive-force-dependent pumps, including RND pumps, that has often increased the antibiotic susceptibility of various MDR bacteria, including *A. baumannii*. In addition, two novel serum-associated EPIs—ABEPI1 and ABEPI2—were identified by [184]. They potentiate the activities of minocycline and ciprofloxacin toward serum-grown *A. baumannii*. Importantly, none of the compounds showed human cytotoxicity, which has limited EPIs development in the past. The clinical translation of these promising EPIs remains currently problematic: off-target effects, relatively low potency, poor pharmacokinetics/pharmacodynamics, and human cell toxicity need to be overcome before advancing EPIs to the clinic for the treating of biofilm-related infections.

#### 5.1.5. New Formulations of Antibiotics

Given the increasing prevalence of antibiotic resistance, the search for new formulations of antibiotics is also of great interest, especially in the case of “last choice” antibiotics. Antibiotic encapsulation and nanoformulation can overcome limitations due to poor drug solubility, stability, and low permeation through biological barriers, such as biofilm. Scutera et al. recently proposed a new formulation comprising chitosan-coated human albumin NPs loaded with colistin (Col/haNPs) against a biofilm formed by MDR strains of *A. baumannii* [154]. Col/haNPs showed relevant biofilm inhibition against both colistin-susceptible and -resistant *A. baumannii* strains and significantly reduced the biofilm also at sub-MICs. By contrast, Col free could reduce biofilm formed by Col-R *A. baumannii* only at higher concentrations.

#### 5.1.6. Antibiofilm Antibodies

Antibodies have also shown excellent therapeutic potential as an adjunct to standard-of-care antibiotics for biofilm-related *A. baumannii* infections. Xiong et al. tested the in vivo antibiofilm efficacy of TRL1068, a high-affinity human monoclonal antibody against an epitope of DNABII proteins that stabilizes biofilm extracellular DNA both in Gram-positive and -negative bacterial species [185]. Treatment with imipenem in combination with TRL1068 caused a significant reduction of *A. baumannii* adhesion to a catheter implanted subcutaneously into mice, higher than that observed after exposure to the antibiotic alone [185]. Another study found that antibodies against the N-terminal domain of CsuE pili completely blocked biofilm formation by *A. baumannii* on hydrophobic plastics, thus warranting further studies aimed at translating this mechanism into a treatment option [43].

#### 5.1.7. Carbon Monoxide Releasing Molecules (CORMs)

CORMs, known as anti-inflammatory and antiapoptotic agents, have recently attracted substantial interest in microbiology due to their antimicrobial activity. Phenanthroline-based manganese(I) photoCORMs inhibited the formation of *A. baumannii* biofilms at concentrations not toxic to *Galleria mellonella* larvae; spectrophotometric findings suggested an antibacterial mode of action due to the combination of CO release as well as the production of photo byproducts released after photoCORM irradiation with visible light (blue or green) [162]. Thus, these compounds may have high translational applications, particularly as surface and medical device sterilizing agents, as well as in the treatment of exposed infected wounds, where a photolyzing source may access the sites of application.

#### 5.1.8. Iron Chelation

A number of environmental factors can influence biofilm formation, including the presence of metal cations, such as iron, an essential nutrient for infecting bacteria, and a key determinant in host−pathogen interactions. In particular, both siderophore production and motility-related biofilm formation were found modulated by iron levels in *A. baumannii* [186]. A novel and effective antibiofilm compound maipomycin A, isolated from the metabolites of the marine actinomycete strain *Kibdelosporangium phytohabitans* XY-R10, was recently described by [163]. Maipomycin A inhibits *A. baumannii* from forming biofilms on medical materials in vitro, such as catheters (silicone) and endotracheal tubes (polyvinyl chloride), partially due to iron chelation.

#### 5.1.9. Antipersister Effects

Persister cells have been implicated in biofilm tolerance to antibiotics. Indeed, a subset of cells in a biofilm are dormant and non-growing; thus, in contrast to actively growing cells, these cells persist in spite of antibiotic treatment and are aptly termed ”persister” [14]. Antipersister activity might, therefore, represent a promising target for the development of new antibiofilm strategies. In this regard, ZY4—a cyclic peptide designed on cathelicidin-BF15 and stabilized by a disulfide bridge—was demonstrated to inhibit biofilm formation by *P. aeruginosa* and *A. baumannii* and exhibited biofilm eradication activity secondary to killing the dormant persister cells by permeabilizing the bacterial membrane [164].

### 5.2. Disrupting Preformed Biofilm

The dispersal of mature biofilm is an important strategy for controlling biofilms. The potential strategies currently under study to disrupt preformed biofilms by *A. baumannii* are summarized in Table 2.

#### 5.2.1. Peptides

Scientists have gradually turned to peptides that are able to eliminate preformed biofilms [194]. Cationic peptides, especially those enriched in lysine and arginine, were considered to exhibit favorable antibiofilm potential because of their affinity to the biofilm surface and the resulting degradation of established biofilms. The cationic amphiphilic peptide zp3 (GIIAGIIIKIKK-NH2) was shown to significantly affect, in a dose-dependent manner, both biofilm formation and mature biofilm [155]. Circular dichroism showed that zp3, similarly to other peptides with a helical structure, destabilizes cell membranes leading to pore formation and finally causes the biofilm architecture to collapse. It also binds and aggregates on the surface of preformed biofilm, thus resulting in its partial dissolution, while maintaining low cytotoxicity to mammalian cells [155]. In another study, it has been observed that Pro10-1D, a novel short peptide designed from insect defensin, inhibited biofilm formation and disrupted the mature biofilms formed by MDR *A. baumannii* strains [156]. Examination by optic microscope indicated that direct targeting of cell membrane and EPS disruption might be the possible mechanisms responsible for its potent, concentration-dependent, antibiofilm activity. Liu et al. found that cecropin antimicrobial peptide Cec4 at 1 × MIC (4 μg/mL) could clear more than 20% of mature biofilms, with a minimal biofilm eradication concentration (MBEC)-50 of 16 μg/mL and MBEC-80 of 128 μg/mL [143]. SEM analysis performed on biofilm formed on urinary catheters confirmed the ability of Cec4 at 1 × MIC to destroy the biofilm structure, consisting of loosely distributed cells, most of them with signs of collapse and disintegration.

Ceragenins (CSAs), synthetic mimics of AMPs, were recently developed as a new antibiotic class. In addition to their efficacy against a broad spectrum of microorganisms, causing the formation of transient pores in the membrane resulting in membrane depolarization and cell death, CSAs are preferable to AMPs due to their higher stability in host tissues and lower production costs [195]. CSAs are also effective against polymicrobial biofilms. In this regard, CSA-13 was found to be active against *A. baumannii*—*Candida* spp. mixed biofilms, as also confirmed through fluorescence microscopy experiments; contrarily, AMPs were found ineffective against multispecies biofilm [187].

Increasing evidence indicates that indole and its derivatives—relevant in the pharmaceutical, agricultural, chemical, and material sciences—exhibit antimicrobial and antibiofilm activities against MDR bacteria [196]. Of the 24 indole derivatives examined, Raorane et al. found four potently inhibited *A. baumannii* biofilm formations in vitro [157]. In particular, 5-iodoindole effectively inhibited biofilm formation and growth and promoted biofilm dispersal, as confirmed by confocal microscopy.

#### 5.2.2. Photodynamic Inactivation

Photodynamic inactivation (PDI) causes a rapid killing of microbial cells, utilizing the combination of a nontoxic dye known as photosensitizer (PS) and visible light to produce cytotoxic reactive oxygen species that can damage cellular components (e.g., DNA, membrane lipids and proteins), ultimately leading to cell death. To achieve an efficient PDI, the PS must bind to or penetrate the bacterial cell. PSs bearing an anionic charge, such as erythrosine B, need agents that increase their penetration into the OM of Gram-negative bacteria. The conjugation of PSs with a variety of NPs increases the efficiency of PSs. Chitosan nanoparticles have some advantages, such as being nontoxic, low immunogenic and biodegradable. In this regard, Pourhajibagher et al. observed that, using a diode laser as a light source in a wavelength of 810 nm exposure, the PS indocyanine green encapsulated in chitosan NPs caused significant disruption to preformed biofilms by *A. baumannii* isolates from burn wounds, as also confirmed by SEM analysis [158]. In another study, Fekrirad et al. found that chitosan at 1/2× MIC enhanced the antibiofilm efficacy of erythrosine-mediated-PDI against *A. baumannii* preformed biofilm causing a marked decrease in the number of viable biofilm cells (>3 log10 colony forming units, CFU/mL) [188]. In addition, methylene blue and protoporphyrin IX were effectively used as PSs, using a halogen lamp as a light source, against *A. baumannii* biofilms. Methylene blue showed a higher bacterial reduction of CFU than protoporphyrin IX (7.0 vs. 6.0 log_10_, respectively) [189].

Silver sulfadiazine (AgSD) is frequently used for controlling microbial infection, especially in burn wounds. Due to AgSD local toxicity, a nanoliposomal (NL) form has been proposed to increase the benefit/risk ratio. To enhance the effectiveness of this treatment method for burn wound infections, the AgSD-NL combined with curcumin as a PS was used in the PDI process by [159]. Photoexcited AgSD-NLs@Cur showed antibacterial and antibiofilm activity against *A. baumannii* involved in burn wound infections, without any significant cell cytotoxicity and hemolytic activity. This effect was likely secondary to the downregulation of the *abaI* gene that in *A. baumannii* is involved in AHL production and biofilm formation [159].

#### 5.2.3. Phage-Based Therapy

Several in vivo studies have recently shown that bacteriophages—i.e., viruses that specifically infect bacteria without damaging eukaryotic cells—might represent a potential alternative to antibiotics for the treatment of *A. baumannii* infections [197,198]. Phages and phage-associated enzymes have been extensively studied as strategies for biofilm prevention and eradication. The phage ISTD, isolated from Belgrade wastewaters, yielded a 2-log reduction in *A. baumannii* biofilm-associated viable bacterial cell count [190]. However, the effect was time-dependent, thus indicating that phage treatment provides a window in which bacteria might be eradicated by some other antimicrobial agent, presumably another phage, or an antibiotic.

Indeed, another strategy might be to enhance bacteriophage activity by combining them with antibiotics. Such a combination was demonstrated to be synergistic to control the emergence of *A. baumannii* resistant strains [199]. Grygorcewicz et al., found that ciprofloxacin, cotrimoxazole, gentamicin, tobramycin or imipenem combined with an environmental bacteriophage, resulted in a significant, dose-dependent, reduction of biofilm biomass formed by uropathogenic *A. baumannii* under human urine, as well as in the clearance of persister cells [191].

Endolysins are phage-encoded enzymes able to degrade the bacterial cell walls at the end of a phage replication cycle, and therefore have been investigated as therapeutic agents [200]. They have had effective results against a wide range of Gram-positive bacteria, although Gram-negative bacteria are commonly not susceptible, due to an OM that acts as a physical barrier [201]. Despite that, Yuan et al. recently observed that endolysin Abtn-4, purified from bacteriophage D2 isolated from hospital wastewater, significantly reduced the viability of biofilms by *A. baumannii* strains causing pneumonia [160]. SEM micrographs confirmed that exposure to Abtn-4 caused a clear reduction in biofilm cell densities. Abtn-4 acts as a glycoside hydrolase against bacterial peptidoglycan layers and contains an amphipathic helix that probably broadens its bactericidal spectrum. 

#### 5.2.4. Potentiation of Antibiotics

Much attention has been recently directed to the potentiation of antibiotics as a new approach against biofilm-related infections, although information available is largely not specific to *A. baumannii*. The evidence of synergism between QSIs and antibiotics against resistant pathogens [202] might represent a promising therapeutic strategy. In this regard, Bhattacharya et al. found that ciprofloxacin combined with ursolic acid, and doxycycline with betulinic acid significantly reduced the viability of *A. baumannii* biofilm cells, likely due to the increase in antibiotic diffusion mediated by the triterpenoids [149].

Recently, Peng et al. assessed whether five biofilm inhibitors (i.e., zinc lactate, stannous fluoride, furanone, azithromycin, and rifampicin) could act as “potentiators” of four conventional anti-*A. baumannii* antibiotics (imipenem, meropenem, tigecycline or polymyxin B) against biofilm formed by extensively drug-resistant *A. baumannii* [161]. Synergism was detected only when stannous fluoride was used with imipenem, meropenem, and tigecycline, respectively in 33%, 44%, and 56% of the isolates. Although stannous fluoride cannot be administered systematically due to toxicity concerns, its inhibitory concentrations can be readily reached in topical use as happens in oral care formulations, thus suggesting the potential to combat wound infections associated with *A. baumannii*.

#### 5.2.5. Nanoparticles

Due to their peculiar characteristics, NPs are considered important candidates in the development of antibacterial and antibiofilm agents. The large surface area-to-volume ratio ensures indeed a wide range of reactions when they meet microbial surfaces [203]. Furthermore, the development of microbial resistance is very unlikely as NPs have multiple targets, and microorganisms would have to undergo a series of mutations simultaneously [204]. In particular, Al_2_O_3_ NPs exert antibiotic activity due to their positive charge that can easily interact with negatively charged bacterial surfaces, reactive oxygen species generation, and self-promoted uptake mechanism, resulting in cell death [205]. Al_2_O_3_ NPs at subinhibitory concentrations affected biofilm formation and pre-established *A. baumannii* biofilms, although these effects were less over time, due to a decreased NP affinity for treated cells [152]. An alternative to the physicochemical synthesized NPs might be the mycofabrication of silver nanoparticles (AgNPs) through a green chemistry protocol. Neethu et al. showed that the surface functionalization of a central venous catheter with mycogenic AgNPs, synthesized using the filtrate of marine fungus *Penicillium polonicum* ARA10, was highly effective against biofilm formation by highly pathogenic *A. baumannii*, as evidenced by SEM and field emission-SEM images [192].

#### 5.2.6. Hydrogel-Based Formulations

The compound CZ-01179, belonging to a broader class of newly synthesized antibiofilm agents and formulated as the active agent in a hydrogel, was tested for its efficacy and ability to treat and prevent biofilm-related wound infection caused by *A. baumannii*, both in vitro and in vivo. Compared to a clinical-standard silver, sulfadiazine-CZ-01179 was significantly more effective at eradicating biofilms in vitro and up to 6 days faster at eradicating biofilms in vivo in a pig excision wound model [193]. In another work, Lopez-Carrizales et al. generated novel chitosan hydrogels (CHs) loaded with AgNPs and ampicillin (AMP) to prevent the early formation of biofilms on central venous catheters [165]. CHs containing 25 ppm of AgNPs and 50 ppm AMP dramatically inhibited (10-log_10_ reduction) the formation of biofilms of MDR *A. baumannii*, *E. faecium* and *S. epidermidis*.

## 6. Conclusions and Future Perspectives

Biofilms are considered to dominate and shape life on Earth. Interestingly, the human gut, skin and oral microbiota comprise bacterial cells organized in complex biofilms. However, biofilms produced by pathogens such as *A. baumannii* in those environments in which human beings are most vulnerable (e.g., hospitals and communities) represent an increasing threat to our health. For this reason, studies of this microorganism have been boosted in the last decade. Some typical features have emerged; an increasing body of evidence highlights the high degree of heterogeneity among healthcare-associated isolates differentiating the ones tightly connected with the host (patients) vs. those strains residing in hospital settings. We are learning that *A. baumannii* is a master of adaptation. It easily adapts to whatever stimuli it perceives from the environment, with the ability to resist desiccation, antibiotics, disinfectants, and the host’s immune defenses. In the last decades, *A. baumannii* MDR strains have thrived in healthcare environments and the biofilm-forming activity is a successful strategy that *A. baumannii* uses for survival and persistence. Several promising approaches are under investigation to prevent biofilm formation within hospital settings, including the downregulation of biofilm-associated genes, inhibition of QS signals, EPS production, efflux pumps as well as new antibiotics and antibiofilm antibodies. In addition, although once established, *A. baumannii* biofilms are very difficult to treat; ongoing new strategies are aimed at disrupting preformed biofilm, such as the use of peptides, photodynamic inactivation, phage-based therapy, and potentiation of antibiotics. Currently, as biofilms are such a public health issue, well-trained health care workers should be more closely involved in the detection, management, and treatment of healthcare-associated biofilms. In the near future, it would be highly desirable to see a synergic effort from scientists, clinicians, and pharmaceutical companies to effectively eradicate biofilms.

## Figures and Tables

**Figure 1 microorganisms-09-01353-f001:**
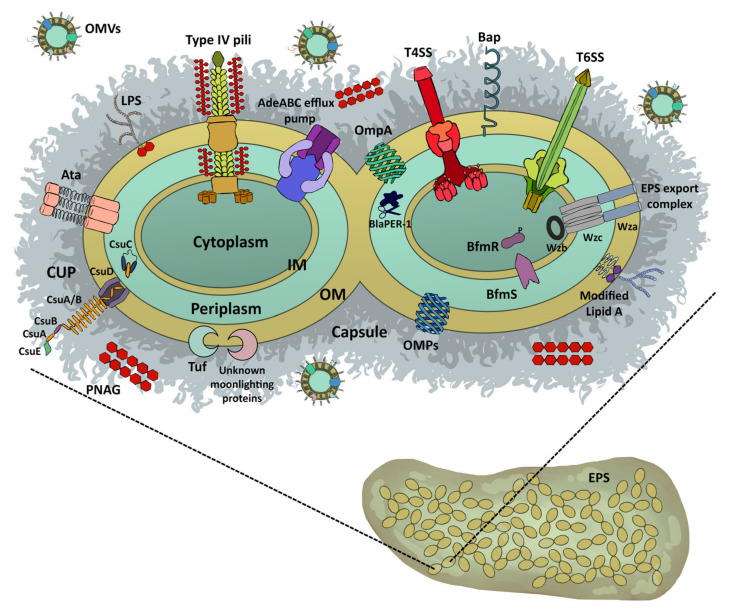
*A. baumannii* surface structures involved in biofilm formation and maintenance. The picture shows a magnification of *A. baumannii* forming a biofilm with the bacterial cells embedded in the extracellular polymeric substance. See the text for details. AdeABC, RND efflux pump; Ata, autotransporter; Bap, biofilm-associated protein; BfmRS, two-component system; BlaPER-1, extended-spectrum β-lactamase; CUP, chaperone-usher pili; EPS, extracellular polymeric substance; IM, inner membrane; LPS, lipopolysaccharide; OM, outer membrane; OMPs, outer membrane proteins; OMVs, outer membrane vesicles; PNAG, poly-β-(1-6)-N-acetylglucosamine; T4SS, type IV secretion system; T6SS, type VI secretion system; Tuf, moonlighting protein; Wza-Wzb-Wzc system.

**Table 1 microorganisms-09-01353-t001:** Strategies aimed at preventing biofilm formation by *A. baumannii*.

Strategy	Strains/Isolates	Antibiofilm Activity	Antibiofilm Mechanisms	Reference
Pyrogallol (polyphenolic organic compound from galls and barks of various trees)	MTCC 9829 reference strain	MBIC: 20 μg/mL	Reduced CSH Reduced motility (swarming) Reduced EPS production Downregulation of adhesion-related genes (*ompA*, *csuA*/*B*) Downregulation of *bap* gene involved in biofilm formation and stabilization	[138]
Myrtenol (bicyclic monoterpene derived from various plants)	ATCC 19606 and MTCC 9826 reference strains Clinical isolates (*n* = 2)	MBIC: 200 μg/mL CLSM showed reduced biomass, maximum thickness, and surface-to-volume ratio	Reduced CSH Reduced motility (swarming, twitching) Downregulation of biofilm-associated genes (*bfmR*, *csuA/B*, *bap*, *ompA*, *pgaA*, *pgaC*)	[139]
Curcumin (flavonoid)	*A. baumannii* ATCC 17,978 reference strainClinical isolates (*n* = 8)	Biofilm inhibition by: 46% at 20 μg/mL 93% at 100 μ/mL	Reduced motility (swimming, swarming) Interaction with the biofilm response regulator BfmR	[140,141]
5-hydroxymethylfurfural (furan organic compound derived from the dehydration of reducing sugars)	ATCC 19606 reference strain	MBIC: 100 μg/mL	Reduced CSH Reduced motility (swarming, twitching) Reduced EPS production Downregulation of biofilm-related genes (*bap*, *csuA/B*, *ompA*, *bfmR*, *katE*)	[142]
Synthetic peptide Cec4	Carbapenem-resistant isolates (*n* = 200)	MBIC: 64–128 µg/mL	Reduced motility (twitching) Downregulation of biofilm-related genes (*csuE*, *bfmR and bfmS*, *bap*)	[143]
CFS from *Clostridium butyricum*	ATCC 19606 reference strain MDR clinical isolates (*n* = 2)	Biofilm inhibition by: 24.4–33.9%, at 12.5% CFS 28.2–43.1%, at 25% CFS 93.6–99.6%, at 50% CFS	Reduced motility Downregulation of RND-type efflux pump-related *adeABC* genes	[144]
Non-native AbaR antagonists	M2 *abaI::lacZ* (Δ*abaI* reporter) and M2 wild-type	Biofilm inhibition by 40%	QS inhibition	[145]
Siphonocholin (from marine sponge *Siphonochalina siphonella*)	ATCC BAA747 reference strain	Biofilm inhibition by 70%	QS inhibition Reduced motility (swarming) Reduced EPS production	[146]
Flavonoid-rich active fraction F1 (from *Glycyrrhiza glabra*)	ATCC 19,606 and ATCC 17,978 reference strains Clinical isolates (*n* = 5)	Concentration-dependent effect Maximum biofilm inhibition by 30–70%, at 2 mg/mL	QS inhibition by *abaI* downregulation Reduced motility (twitching)	[147]
Linalool (oil compounds from *Coriandrum sativum*)	LMG 1025 and LMG 1041 reference strains Clinical isolates (*n* = 3)	Concentration-dependent effect: 1–18%, at 0.25 × MIC 75–97.1%, at 4 × MIC	Reduced adhesion QS inhibition	[148]
Pentacyclic triterpenoids (betulinic acid, glycyrrhetinic acid, ursolic acid)	ATCC 19606 reference strain	Biofilm inhibition (respectively at 50, 100 and 200 µg/mL): 36, 56, 80% (glycyrrhetinic acid) 31, 63, 88% (ursolic acid) 45, 62, 88% (betulinic acid)	QS inhibition (at AHL synthase and AHL dependent transcriptional activator) Reduced EPS production	[149]
MomL (AHL lactonase belonging to the metallo-β-lactamase superfamily)	LMG10520, LMG10531 and AB5075 reference strains	Concentration-dependent effect Maximum biofilm inhibition by 42% at 5 µg/mL	AHL degrading activity	[119]
Purified QQ enzyme Aii20J	ATCC17978 reference strain MDR clinical strains (*n* = 5)	Biofilm inhibition by 80% The effect is strain-dependent and improved when QQ enzyme is combined with DNase	Decreased the number of surface short pili	[150]
Palmitoleic acid, myristoleic acid (unsaturated fatty acids)	ATCC17978 reference strain Clinical isolates (*n* = 22)	Biofilm inhibition (at 0.02 and 0.05 mg/mL, respectively) by: 37 and 39% (palmitoleic acid) 28 and 42% (mirystoleic acid) Significant biofilm reduction in: 13 isolates (palmitoleic acid) 8 isolates (mirystoleic acid)	Inhibition of *abaR* gene expressionAccumulation of fatty acids at the air−liquid interface, due to their amphiphilic nature	[151]
Al_2_O_3_ synthetic NPs	MDR strains (*n* = 3)	Biofilm inhibition by 11.6 to 70.2% at 0.5xMIC	Reduced EPS production	[152]
Labetalol hydrochloride (Wzb-Wzc interaction inhibitor)	RS 307 reference strain	MBIC: 1 mM	Reduced EPS production	[153]
Chitosan-coated human albumin nanoparticles for the delivery of colistin (Col/haNPs)	ATCC 19,606 reference strain Colistin-susceptible (*n* = 1) and -resistant (*n* = 3) clinical isolates	Significant biofilm inhibition at 1/2x and 1/4xMIC Col/haNPs > 4–60-fold vs. free colistin	Positively charged NPs might adsorb and accumulate on the negatively charged bacterial surface and EPS by electrostatic interactions Prolonged release of colistin Chitosan−colistin synergistic effect	[154]
Polyclonal antibodies vs. self-complemented CsuA/B subunit (αA/B) and CsuE_NTD_ (αE_N_)	Clinical strains (*n* = 5)	αE_N_ inhibits biofilm formation more efficiently than αA/B: αEN diluted up to several thousand times completely blocked biofilm formation αA/B inhibited biofilm formation only at high concentration	Inhibition of the binding to hydrophobic plastics by blocking the three hydrophobic fingers at the tip of CsuE N-terminal domain (CsuE_NTD_)	[43]
Cationic amphiphilic peptide zp3 (GIIAGIIIKIKK-NH_2_)	ATCC 19606 reference strain	Biofilm inhibition by: 20%, at 0.5 μM 100%, at >4 μM	Destabilization of cell membranes with pore formation and consequent biofilm collapse	[155]
Pro10-1D (a short peptide from insect defensin)	KCCM 40203, CCARM 12010 and CCARM 12220 reference strains	Concentration-dependent effect: 20% inhibition at 2 µM >99.9% inhibition at 64 µM	Reduced EPS production	[156]
24 indole derivatives (including 16 halogenated indoles)	ATCC 17978 and ATCC BAA-1709 reference strains MDR clinical isolates (*n* = 7)	Biofilm inhibition at 50 µg/mL: 62% (4-bromoindole) 75% (4-chloroindole) 60% (4-iodoindole) 94% (6-iodoindole) 96% (5-iodoindole)	Reduced surface motility Induced reactive oxygen species, resulting in loss of cell membrane integrity and cell shrinkage	[157]
PDI mediated by indocyanine green encapsulated in chitosan nanoparticles (NCs@ICG-aPDT)	Isolates from burn wounds (*n* = 50)	Biofilm inhibition by: 55.3% after exposure to NCs@ICG-aPDT No inhibition after exposure to NCs@ICG, ICG, and the diode laser alone	Bactericidal effect	[158]
PDI of nanoliposomal silver sulfadiazine doped with curcumin (AgSD-NLs@Cur)	Isolates from burn wounds (*n* = 100)	Biofilm inhibition by 76.4% after exposure to AgSD-NLs@Cur at MIC_90_ and light-emitting diode Photoexcited AgSD and AgSD-NLs at MIC_90_ are more effective than either group without LED irradiation (38.1 vs. 44.8%, respectively)	Ag in AgSD-NLs interacts with DNA and sulfhydryl groups of microbial enzymes, leading to bacterial growth inhibition Downregulation of *luxI* gene	[159]
Endolysin Abtn-4 from phage vB_AbaP_D2 (isolated from hospital wastewater)	Clinical MDR isolates (*n* = 15) AB9 host strain	Biofilm inhibition > 30% following exposure in the early (12 h post-incubation) or pre-maturation phase (36 h post-incubation)	EPS disruption Bacterial cell wall degradation	[160]
Pentacyclic triterpenoids, (glycyrrhetinic acid, ursolic acid, betulinic acid) combined with a conventional antibiotic (doxycycline, roxithromycin or ciprofloxacin)	ATCC 19606 reference strain	Glycyrrhetinic acid and betulinic acid increase antibiofilm activity of doxycycline and roxithromycin Ursolic acid improves the effect of ciprofloxacin	Increased antibiotic diffusion through biofilm mediated by the triterpenoids	[149]
Biofilm inhibitors (zinc lactate, stannous fluoride, furanone, AZM, and RIF) combined with a conventional antibiotic (IMP, MRP, TIG, POL)	XDR clinical isolates (*n* = 9)	Biofilm inhibition: 16 to 50%, at sub-MICs lactate > stannous fluoride > furanone > RIF > AZM Synergistic effects of: zinc lactate, stannous fluoride and furanone combined with TIG (22, 56 and 11% of the isolates, respectively) zinc lactate and stannous fluoride each used with a carbapenem (IMP or MRP), in 33% of the isolates	Zinc compounds inhibit EPS synthesis and the formation of matrix networks Stannous fluoride destroys the biofilm structure by loosening the structure of the biofilm matrix Furanone replaces the binding sites of QS signal molecules Azithromycin inhibits EPS production, leading to the formation of channels that favor antibiotic diffusion through the biofilm	[161]
Phenanthroline-based visible-light-activated manganese (I) carbon-monoxide-releasing molecules (PhotoCORMs)	ATCC BAA 1710 and ATCC 17978 reference strains	Biofilm formation inhibition only at high concentrations (>128 mg/mL) in the dark Compounds 1-2 reveal remarkable activity at 4–8 mg/mL when irradiated with blue LED light Compound 2 shows higher activity than ciprofloxacin vs. MDR ATCC BAA 1710 strain	Antibacterial activity due to the combination of CO release as well as the production of photo-byproducts	[162]
Maipomycin A (from the metabolites of the marine actinomycete *Kibdelosporangium phytohabitans* XY-R10)	ATCC 19606 reference strain	Biofilm inhibition by 84.3% at MBIC (8 μg/mL) Concentration-dependent effect Inhibition of biofilm formed on medical materials, such as catheters (silicone) and endotracheal tubes (polyvinyl chloride)	Fe(II) and Fe(III) ions chelation The chelation of Maipomycin A and iron ions may be negatively affected by other metal as competitors	[163]
ZY4 cyclic synthetic peptide (designed on cathelicidin-BF15 and stabilized by a disulfide bridge)	ATCC 22933 reference strain MDR clinical isolates (*n* = 5)	Concentration-dependent effect 22%, at 0.5xMIC 46%, at 2xMIC 66%, at 8xMIC	Bactericidal effect by permeabilizing the cell membrane	[164]
Chitosan hydrogels loaded with AgNPs and AMP	Carbapenem-resistant isolate from CVC	Biofilm viability inhibition on CVC: 1 log_10_ (chitosan) 10 log_10_ (chitosan with 25 ppm AgNPs and 50 ppm AMP)	NS	[165]

MBIC, minimum concentration of drug that exhibits greater than 50% of biofilm inhibition without affecting growth; CSH, cell surface hydrophobicity; EPS, extracellular polymeric substance; CLSM, confocal laser scanning microscopy; CFS, cell-free supernatant; MDR, multidrug resistant; QS, quorum sensing; MIC, minimum inhibitory concentration; AHL, N-acyl-homoserine lactone; QQ, quorum quenching; NPs, nanoparticles; PDI, photodynamic inactivation; AZM, azithromycin; RIF, rifampicin; XDR, extensively drug resistant; IMP, imipenem; MRP, meropenem; TIG, tigecycline; POL, polymyxin B; AMP, ampicillin; CVC, central venous catheter; NS, not specified.

**Table 2 microorganisms-09-01353-t002:** Strategies aimed at disrupting preformed biofilm by *A. baumannii*.

Strategy	Strains/Isolates	Antibiofilm Activity	Antibiofilm Mechanisms	Reference
Myrtenol (bicyclic monoterpene derived from various plants)	ATCC-19606 and MTCC-9826 reference strains Clinical isolates (*n* = 2)	Significant biofilm dispersion at MBIC for all strains tested	Reduced CSH Reduced motility (swarming, twitching) Downregulation of biofilm-associated genes (*bfmR*, *csuA/B*, *bap*, *ompA*, *pgaA*, *pgaC*)	[139]
Synthetic peptide Cec4	Carbapenem-resistant isolates (*n* = 200)	Biofilm dispersion > 20% at 1xMIC (4 µg/mL) MBEC_50_: 16 µg/mL MBEC_80_: 128 µg/mL MBEC: 256–512 µg/mL	Reduced motility (twitching) Downregulation of biofilm-related genes (*csuE*, *bfmR and bfmS*, *bap*)	[143]
CFS from *Clostridium butyricum*	ATCC 19,606 reference strain MDR clinical isolates (*n* = 2)	Biofilm dispersion by: 24–63.2%, at 12.5% CFS 28.4–84%, at 25% CFS 80.5–92.6%, at 50% CFS Decreased metabolic activity by 92.9–100% at 50% CFS	Reduced motility Downregulation of RND-type efflux pump-related *adeABC* genes	[144]
Linalool (oil compounds from *Coriandrum sativum*)	LMG 1025 and LMG 1041 reference strains Clinical isolates (*n* = 3)	Concentration-dependent biofilm dispersion: 1–28%, at 0.25xMIC 55–86%, at 4xMIC	Reduced adhesion QS inhibition	[148]
Pentacyclic triterpenoids (betulinic acid, glycyrrhetinic acid, ursolic acid)	ATCC 19606 reference strain	Reduction of biofilm viability at 200 µg/mL by: 26% (glycyrrhetinic acid) 23% (ursolic acid) 23% (betulinic acid)	QS inhibition (at AHL synthase and AHL dependent transcriptional activator) Reduced EPS production	[149]
PDI mediated by indocyanine green	ATCC 19606 reference strain	Biofilm viability reduction by 3 log_10_ at 1000 µg/mL (laser irradiation for 1 min, with an estimated average output light energy 31.2 J/cm^2^)	QS inhibition by *abaI* downregulation	[177]
Thermostable QQ phosphotriesterase-like lactonase (from *Geobacillus kaustophilus*)	Clinical isolate	Biofilm dispersion by 75% CLSM confirms reduction in biofilm biomass, thickness, and surface area	Hydrolysis of two biologically relevant C-3-hydroxylated AHLs (3-OH-C_10_-HSL, 3-OH-C_12_-HSL)	[179]
Palmitoleic acid, myristoleic acid (unsaturated fatty acids)	ATCC 17978 reference strain Clinical isolates (*n* = 22)	Biofilm dispersion by 24% for both fatty acids at 0.05 mg/mL AFM shows merging microcolonies with cell elongation	Inhibition of *abaR* gene expression Accumulation of fatty acids at the air−liquid interface, due to their amphiphilic nature	[151]
Al_2_O_3_ synthetic NPs	MDR strains (*n* = 3)	Biofilm dispersion by 11.4 to 56.8% at 0.5xMIC Efficacy reduced over time, due to the loss of affinity of the NPs with treated cells: 11.4 to 20.9% reduction of 120 h-old biofilm 33.8 to 56.8% reduction of 48 h-old biofilm	Reduced EPS production	[152]
TRL1068 (native human monoclonal antibody)	MDR clinical isolate	Biofilm disruption at 1.2 µg/mL for 12 h Murine skin and soft tissue infection model: imipenem + TRL1068 > imipenem + isotype control (reduction by 0.8 and 0.5 log_10_ CFU/catheter, respectively)	Binding to the biofilm component DNABII protein family	[185]
Cationic amphiphilic peptide zp3 (GIIAGIIIKIKK-NH_2_)	ATCC 19606 reference strain	Concentration-dependent effect: 30% at 2 μM (0.5xMIC) >50% at 64 μM (16xMIC)	Destabilization of cell membranes with pore formation and consequent biofilm collapse	[155]
Ceragenins (CSA-13, CSA-44, CSA-90, CSA-131, CSA-138, CSA-142, CSA-144, and CSA-192) synthesized from a cholic acid scaffold technique	ATCC 19606 reference strain	CSA-13 is the most effective, causing 1 log_10_ reduction of biofilm viability at 100 µg/mL All but CSA-144 are effective vs. *A. baumannii-C. albicans* mixed biofilms	NS	[187]
Indole derivatives (including halogenated indoles)	ATCC 17,978 and ATCC BAA-1709 reference strains MDR clinical isolates (*n* = 7)	5-iodoindole at 250 µg/mL causes: 62% biofilm dispersion 95% biofilm killing CLSM shows that 5-iodoindole significantly reduces biofilm biomass, thickness, and substrate coverage	Reduced surface motility Induced reactive oxygen species, resulting in loss of cell membrane integrity and cell shrinkage	[157]
PDI mediated by erythrosine B	ATCC BAA 747 reference strain Isolates from burn wounds (*n* = 2)	PDI mediated by erythrosine B + acetic acid does not cause lethal effect (≥3 log_10_ reduction in CFU) on biofilm A lethal effect is observed after adding chitosan at 1/2xMIC to erythrosine B	Chitosan improves the antibiofilm efficacy of erythrosine B-mediated PDI by: disruption of biofilm structure, thus helping the PDI to act on the cells released from the biofilm permeabilization of the bacterial outer membrane acting as a drug carrier for delivery of erythrosine to biofilm	[188]
PDI mediated by methylene blue and protoporphyrin IX	MDR isolates from clinical (*n* = 1), abattoir (*n* = 1) and aquatic (*n* = 1) sources	Dose-dependent biofilm viability reduction by: 6 log_10_ for protoporphyrin IX 7 log_10_ for methylene blue	Cationic methylene blue is attracted to the negative EPS and cell walls, thus increasing its intracellular concentration following antimicrobial PDI; this causes biofilm detachment secondary to disintegration of the interaction between bacteria, due to reactive oxygen species exposure The negatively charged protoporphyrin IX is electrostatically repulsed from EPS and also from the lipopolysaccharide layer of the bacterial cell wall	[189]
ISTD depolymerase producing phage, isolated from wastewaters	Clinical isolates (89 carbapenem-resistant, 14 carbapenem-sensitive) 6077/12 host strain	Biofilm viability reduction after 6 h exposure to ISTD phages: 0.6 log_10_, at MOI 0.1 2 log_10_, at MOI 100 No reduction after 24 h exposure	NS	[190]
Environmental phages, tested both alone or combined with a conventional antibiotic (CIP, SXT, GN, TOB, IMP, MRP)	MDR isolates (*n* = 25) from urinary tract infections	Biofilm biomass reduction in a human urine model: by 35 to 67.4% after single phage exposure dose-dependent effect of phage cocktail + antibiotic (CIP, SXT, GN, TOB, IMP, or MRP) combination SXT-phage cocktail the most active for biofilm dispersion (94.3 and 98.6% at 1/4× and ½ × MIC, respectively), and bacterial regrowth limitation (to 18% and 6% for 1/4 and ½ × MIC values, respectively)	NS	[191]
Surface functionalization of central venous catheter with mycofabricated silver nanoparticles (AgNPs)	MDR isolate	MBEC: 31.2 μg/mL SEM and FE-SEM analyses confirm eradication of preformed biofilm on the AgNPs-coated surface catheter	The direct interaction of AgNPs to the bacterial cell wall and the subsequent penetration resulted in infiltration of AgNPs through the biofilm structure causing physical dispersion of biofilms and bacterial lysis	[192]
ZY4 cyclic synthetic peptide (designed on cathelicidin-BF15 and stabilized by a disulfide bridge)	ATCC 22933 reference strain MDR clinical isolates (*n* = 5)	Concentration-dependent effect More than half of the persisters eliminated at 1 × MIC	Bactericidal effect by permeabilizing the cell membrane	[164]
Hydrogel formulation of first-in-class series of antibiofilm antibiotic (CZ-01179)	ATCC BAA-1605 reference strain CDC clinical isolates (*n* = 11)	CZ-01179 is more effective than clinical standard (silver sulfadiazine) at eradicating biofilms both in vitro and up to 6 days faster at eradicating biofilms in a pig wound model	NS	[193]

MBIC, minimum concentration of drug that exhibits greater than 50% of biofilm inhibition without affecting growth; CSH, cell surface hydrophobicity; MIC, minimum inhibitory concentration; MBEC, minimum concentration eradicating preformed biofilm; MBEC-50, minimum concentration that kills 50% of cells in preformed biofilm; MBEC-80, minimum concentration that kills 80% of cells in preformed biofilm; CFS, cell-free supernatant; MDR, multidrug resistant; QS, quorum sensing; AHL, N-acyl homoserine lactone; EPS, extracellular polymeric substance; PDI, photodynamic inactivation; QQ, quorum quenching; CLSM, confocal laser scanning microscopy; AFM, atomic force microscopy; NPs, nanoparticles; SEM, scanning electron microscopy; NS, not specified; MOI, multiplicity of infection; CIP, ciprofloxacin; SXT, cotrimoxazole; GN, gentamicin; TOB, tobramycin; IMP, imipenem; MRP, meropenem; FE-SEM, field-emission scanning electron microscopy.

## Data Availability

Not applicable.

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
