# Peer review of "Gram-Negative Bacteria Holding Together in a Biofilm: The Acinetobacter baumannii Way"

_microorganisms, 2021, doi:10.3390/microorganisms9071353_

Round 1
Reviewer 1 Report
The article is well written, the scientific information is presented in a logical and very comprehensive order, the material is easy to follow. Also, the references are adapted according to the text and many of the cited articles are recently published (2020, 2021)
I made only a few recommendations:
Abstract: Line 22-23 - “Nosocomial infections” - I recommend to be replace with “health care associate infection” or “Hospital-Acquired Infection”
Introduction - I recommend to discuss about Acinetobacter as health-care associate pathogen in close relation with ESKAPEE group, also referred to in Chapter 5, item 5.1.4.
- Prevention and treatment strategies against A. baumannii biofilm
What about essential oils as antibiofilm herbal products? There are a few recent papers related with this subject (recommended below) and I consider that it is important subject to be highlighted in your paper.
- Rojita Mishra, Amrita Kumari Panda, Surajit De Mandal, Muhammad Shakeel, Satpal Singh Bisht and Junaid Khan. Natural Anti-biofilm Agents: Strategies to Control Biofilm-Forming Pathogens. Front. Microbiol., 29 October 2020 | https://doi.org/10.3389/fmicb.2020.566325
- Ismail M.M., Samir R., Saber F.R., Ahmed S.R.,Farag M.A. Pimenta oil as a potential treatment for acinetobacter baumannii wound infection: In vitro and in vivo bioassays in relation to its chemical composition. Antibiotics Open Access, Volume 9, Issue 10, Pages 1 – 16, October 2020 Article number 679. DOI 10.3390/antibiotics9100679.
Minor revision:
Lines 333, 340, 495, 500, 501, 512, 539…. “In vivo” and “in vitro” – use italic “In vivo” and “in vitro”
Author Response
We want to thank you very much for your interest and fruitful suggestions. Your valuable comments improved significantly our manuscript. In accordance with your comments, we revised and amended the manuscript accordingly. As requested, the revised manuscript file includes outlines of every change made in response to your comments (visible with the "Track Changes function” on).
Below, you can find the point-by-point replies.
The article is well written, the scientific information is presented in a logical and very comprehensive order, the material is easy to follow. Also, the references are adapted according to the text and many of the cited articles are recently published (2020, 2021)
I made only a few recommendations:
Point 1: Abstract: Line 22-23 - “Nosocomial infections” - I recommend to be replace with “health care associate infection” or “Hospital-Acquired Infection”
Response 1. Yes, “nosocomial infections” was replaced with healthcare-acquired infections” throughout the revised manuscript.
Point 2: Introduction - I recommend to discuss about Acinetobacter as health-care associate pathogen in close relation with ESKAPEE group, also referred to in Chapter 5, item 5.1.4.
Response 2. Yes, a sentence about A. baumannii belonging to the ESKAPEE group was added in the Introduction section of the revised manuscript.
Point 3. 5. Prevention and treatment strategies against A. baumannii biofilm
What about essential oils as antibiofilm herbal products? There are a few recent papers related with this subject (recommended below) and I consider that it is important subject to be highlighted in your paper.
- Rojita Mishra, Amrita Kumari Panda, Surajit De Mandal, Muhammad Shakeel, Satpal Singh Bisht and Junaid Khan. Natural Anti-biofilm Agents: Strategies to Control Biofilm-Forming Pathogens. Front. Microbiol., 29 October 2020 | https://doi.org/10.3389/fmicb.2020.566325
- Ismail M.M., Samir R., Saber F.R., Ahmed S.R.,Farag M.A. Pimenta oil as a potential treatment for acinetobacter baumannii wound infection: In vitro and in vivo bioassays in relation to its chemical composition. Antibiotics Open Access, Volume 9, Issue 10, Pages 1 – 16, October 2020 Article number 679. DOI 10.3390/antibiotics9100679.
Response 3. Yes, thank you so much for highlighting this point. In the revised version of the manuscript, we added both papers suggested by the Reviewer in paragraph 5 and in Table 1. Furthermore, a careful reading of the review by Mishra et al (2020) allowed us to discover a relevant study by Raorane et al (2019) focused on the antibiofilm potential of the flavonoid curcumin, whose findings were added to the revised version of the manuscript. This change made necessary to rephrase the title of the section 5.1.1. as “Modulation of genes involved in biofilm formation”.
Point 4. Minor revision:
Lines 333, 340, 495, 500, 501, 512, 539…. “In vivo” and “in vitro” – use italic “In vivo” and “in vitro”
Response 4. Yes, we apologize for this inaccuracy and we amended them throughout the revised manuscript.
Reviewer 2 Report
Acintobacter is a major problem in institutions associated with health care. One of the reasons is survival in dry conditions, transition to VBNC form, and the creation of biofilm and antibiotic resistance. The question remains whether Acinetobacter forms a biofilm or some other form of bonding in dry conditions. The paper is written in detail but I have some doubts:
can nosocomial be replaced in parts of the text in Healthcare-acquired infections? in section 2 most of the text refers to gram negative bacteria in general and in the end mentions Acinetobacter. Either the title needs to be changed or the text needs to be rearranged. Figure 1 showed all the important structures on the cell that are related to the biofilm creation property so it may be better to be at the end of the text describing all these structures and before the antibiofilm effect. The text below the image is too detailed and should be linked to the text. Check if in vivo, sholud be written in Italic.
The table should be resized to remove the bullets when enumerating or not to center the text.
Check how it is spelled A. baumannii ATCC 19606. I don't think the comma goes between the numbers as you stated.
Author Response
We want to thank you very much for your interest and fruitful suggestions. Your valuable comments improved significantly our manuscript. In accordance with your comments, we revised and amended the manuscript accordingly. As requested, the revised manuscript file includes outlines of every change made in response to your comments (visible with the "Track Changes function” on).
Below, you can find the point-by-point replies.
Acintobacter is a major problem in institutions associated with health care. One of the reasons is survival in dry conditions, transition to VBNC form, and the creation of biofilm and antibiotic resistance. The question remains whether Acinetobacter forms a biofilm or some other form of bonding in dry conditions. The paper is written in detail but I have some doubts:
Point 1. can nosocomial be replaced in parts of the text in Healthcare-acquired infections?
Response 1. Yes, the word nosocomial was replaced with “healthcare-acquired infections” throughout the revised manuscript.
Point 2. in section 2 most of the text refers to gram negative bacteria in general and in the end mentions Acinetobacter. Either the title needs to be changed or the text needs to be rearranged.
Response 2. Yes, following reviewer’s advice, in the revised manuscript, we modified the title and the first sentence of the Abstract accordingly.
Point 3. Figure 1 showed all the important structures on the cell that are related to the biofilm creation property so it may be better to be at the end of the text describing all these structures and before the antibiofilm effect.
Response 3. Yes, Figure 1 was moved below paragraph 3.1 in the revised manuscript.
Point 4. The text below the image is too detailed and should be linked to the text.
Response 4. Yes, the text below Figure 1 was greatly shortened and linked to the text in the revised manuscript.
Point 5. Check if in vivo, sholud be written in Italic.
Response 5. Yes, we apologize for this inaccuracy and we amended them throughout the revised manuscript.
Point 6. The table should be resized to remove the bullets when enumerating or not to center the text.
Response 6. Yes, Tables were resized and bullets were removed in the revised manuscript.
Point 7. Check how it is spelled A. baumannii ATCC 19606. I don't think the comma goes between the numbers as you stated.
Response 7.Yes, we are sorry for the mistyping and we amended them throughout the revised manuscript.